# Directing Generative Networks with Weighted Maximum Mean Discrepancy

## Abstract

The maximum mean discrepancy (MMD) between two probability measures $\mathbb{P}$ and $\mathbb{Q}$ is a metric that is zero if and only if all moments of the two measures are equal, making it an appealing statistic for two-sample tests. Given i.i.d. samples from $\mathbb{P}$ and $\mathbb{Q}$, Gretton et al. (2012) show that we can construct an unbiased estimator for the square of the MMD between the two distributions. If $\mathbb{P}$ is a distribution of interest and $\mathbb{Q}$ is the distribution implied by a generative neural network with stochastic inputs, we can use this estimator to train our neural network. However, in practice we do not always have i.i.d. samples from our target of interest. Data sets often exhibit biases—for example, under-representation of certain demographics—and if we ignore this fact our machine learning algorithms will propagate these biases. Alternatively, it may be useful to assume our data has been gathered via a biased sample selection mechanism in order to manipulate properties of the estimating distribution $\mathbb{Q}$.

In this paper, we construct an estimator for the MMD between $\mathbb{P}$ and $\mathbb{Q}$ when we only have access to $\mathbb{P}$ via some biased sample selection mechanism, and suggest methods for estimating this sample selection mechanism when it is not already known. We show that this estimator can be used to train generative neural networks on a biased data sample, to give a simulator that reverses the effect of that bias.

## 1 Introduction

Neural networks with stochastic input layers can be trained to approximately sample from an arbitrary probability distribution $\mathbb{P}$ based on samples from $\mathbb{P}$ (Goodfellow et al., 2014). Generating simulations from complex distributions has applications in a large number of fields: We can automatically generate illustrations for text (Zhang et al., 2017) or streams of video (Vondrick et al., 2016); we can simulate novel molecular fingerprints to aid scientific exploration (Kadurin et al., 2017); and, we can synthesize medical time-series data that can be shared without violating patient privacy (Esteban et al., 2017).

In this paper, we consider the setting of a feedforward neural network (referred to as the generator) that maps random noise inputs $z \in \mathbb{R}^d$ to some observation space $\mathcal{X}$. The weights of the neural network are trained to minimize some loss function between the resulting simulations and exemplars of real data. The general form of the resulting distribution $\mathbb{Q}$ over simulations $G(z)$ is determined by the architecture of the generator—which governs the form of the mapping $G$—and by the loss function used to train the generator. Generative adversarial networks (Goodfellow et al., 2014) use dynamically varying, adversarially learned loss functions specified in terms of the output of a classifier. Other generative networks use a loss function defined using a distributional distance or divergence between the simulation distribution $\mathbb{Q}$ and a target distribution $\mathbb{P}$ (Arjovsky et al., 2017; Nowozin et al., 2016; Zhao et al., 2017a), requiring the generator to mimic the variance in a collection of data points rather than simply converge to a single mode. In particular, the maximum mean discrepancy (Gretton et al., 2012) has demonstrated good performance as a loss function in this setting (Sutherland et al., 2016; Li et al., 2017; Sutherland et al., 2017), since it reduces to zero if and only if all moments of two distributions are equal, requiring the generator to reproduce the full range of variation found in the data.

These approaches, like most machine learning methods, assume our data is a representative sample from the distribution of interest. If this assumption is correct, minimizing the distributional distance between the simulations and the data is equivalent to learning a distribution that is indistinguishable under an appropriate two-sample test from our target distribution. However, we run into problems if our data is not in fact a representative sample from our target distribution—for instance, if our data gathering mechanism is susceptible to *sample selection bias*. The problem of machine learning algorithms replicating and even magnifying human biases is gathering increasing awareness (Bolukbasi et al., 2016; Zhao et al., 2017b), and if we believe our dataset suffers from such biases—for example, if our audio dataset contains primarily male speakers or our image dataset contains primarily white faces – we will typically want to take steps to correct this bias.

Even if our data is representative of the underlying distribution, we might want to generate samples from a modified version of this distribution. For example, we might want to alter the demographics of characters in a scene to fit a story-line. In this setting, we can treat our desired modified distribution as our target distribution, and treat our data as if they were sampled from this distribution subject to an appropriately biased sample selection mechanism.

If we know the form of our sample selection bias, we can reformulate our loss function to penalize the generator based on the difference between simulated data and the unbiased distribution of interest, which we will refer to as our target distribution. After a review of relevant background information in Section 2, we show in Section 3 that, given a function that describes how our observed data deviates from this target distribution, we can construct an estimator of the MMD between the generator and the target distribution.

In practice, we will not know the function linking the target distribution and the empirical data distribution. However, we can approximate this function based on user-provided examples of data points that are over- and under-represented. In Section 4, we discuss ways to estimate this function, and in Section 5 we discuss related work in survey sampling statistics and bias reduction. We demonstrate the efficacy and applicability of our approach in Section 6.

## 2 BACKGROUND

### 2.1 GENERATIVE NETWORKS

Generative networks are a class of models which take a collection of data lying in some observation space $\mathcal{X}$ as input, and aim to generate simulations that are similar to that data – or more generally, whose empirical distribution is close to the distribution underlying the data. These models do not use or require an explicit probability distribution for the data, but instead rely on the fact that sufficiently complex neural networks have the capacity to approximate any arbitrary function (Hornik et al., 1989). We can therefore construct a method of simulating from an arbitrarily complex probability distribution $\mathbb{Q}$ on $\mathcal{X}$ by using a neural network generator $G : \mathbb{R}^d \to \mathcal{X}$ to transform random $d$-dimensional inputs $z$ into simulations $G(z) \in \mathcal{X}$. In order to minimize the difference between the probability distribution $\mathbb{Q}$ over simulations $G(z)$ and the target distribution $\mathbb{P}$, we train the neural network to minimize a loss function between simulations $G(z) \sim \mathbb{Q}$ and data $x \sim \mathbb{P}$

The most common forms of generative network are generative adversarial networks (GANs, Goodfellow et al., 2014), so called because the loss function is dynamically defined in terms of the output of an adversarially learned classifier. This classifier—itself a neural network—is trained to differentiate between two classes, data and simulations, and for a given observation returns a score under the two classes (loosely corresponding to a probability of belonging to that class). The generator's loss function is a function of the score assigned to simulations under the true data class, so that reducing the loss function leads to an increased chance of fooling the classifier.

While this adversarial framework has the advantage of a dynamically evolving and increasingly discriminative loss function, a disadvantage is that the generator can successfully minimize this loss function by mimicking only a subset of the data, leading to a phenomenon known as mode collapse (Salimans et al., 2016; Che et al., 2017). To avoid this, recent works have incorporated estimators of distributional distances between $\mathbb{P}$ and $\mathbb{Q}$, that consider the overall distributions rather than just element-wise distances between samples. For example, the maximum mean discrepancy (Gretton et al., 2012; Sutherland et al., 2016; Li et al., 2017; Sutherland et al., 2017) between two dis-

tributions, which we explore further in Section 2.2, reduces to zero if and only if all moments of the two distributions are the same. Other distributional distances that have been used include the Wasserstein distance and the Cramer distance (Arjovsky et al., 2017; Bellemare et al., 2017). In some cases, these distance-based generative networks include adversarial components in their loss functions; for example the MMD-GAN of Li et al. (2017) adversarially learns parameters of the maximum mean discrepancy metric.

## 2.2 MAXIMUM MEAN DISCREPANCY BETWEEN TWO DISTRIBUTIONS

The maximum mean discrepancy (MMD, Gretton et al., 2012) projects two distributions $\mathbb{P}$ and $\mathbb{Q}$ into a reproducing kernel Hilbert space (RKHS) $\mathcal{H}$, and looks at the maximum mean distance between the two projections, i.e.

$$\text{MMD}[\mathbb{P}, \mathbb{Q}] := \sup_{f \in \mathcal{H}} \left( \mathbf{E}_{x \sim \mathbb{P}}[f(x)] - \mathbf{E}_{y \sim \mathbb{Q}}[f(y)] \right).$$

If we specify the *kernel mean embedding* $\mu_{\mathbb{P}}$ of $\mathbb{P}$ as $\mu_{\mathbb{P}} = \int k(x, \cdot) d\mathbb{P}(x)$, where $k(\cdot, \cdot)$ is the characteristic kernel defining the RKHS, then we can write the square of this distance as

$$\text{MMD}^2[\mathbb{P}, \mathbb{Q}] = ||\mu_{\mathbb{P}} - \mu_{\mathbb{Q}}||_{\mathcal{H}}^2 = \mathbb{E}_{\mathbb{P}}[k(x, x')] - 2\mathbb{E}_{\mathbb{P}, \mathbb{Q}}[k(x, y)] + \mathbb{E}_{\mathbb{Q}}[k(y, y')].$$

Since we have projected the distributions into an infinite-dimensional space, the distance between the two distributions is zero if and only if all their moments are the same.

In order to be a useful loss function for training a neural network, we must be able to estimate the MMD from data, and also take derivatives of this estimate with respect to the network parameters. We can construct an unbiased estimator of square of the MMD (Gretton et al., 2012) using $m$ samples $x_i \sim \mathbb{P}$ and $n$ samples $y_i \sim \mathbb{Q}$ as

$$\widehat{\text{MMD}}^2[\mathbb{P}, \mathbb{Q}] = \frac{1}{m(m-1)} \sum_{i=1}^{m} \sum_{j \neq i}^{m} k(x_i, x_j) + \frac{1}{n(n-1)} \sum_{i=1}^{n} \sum_{j \neq i}^{n} k(y_i, y_j) \\ - \frac{2}{mn} \sum_{i=1}^{m} \sum_{j=1}^{n} k(x_i, y_j). \tag{1}$$

If $\mathbb{P}$ is the distribution underlying our data and $\mathbb{Q}$ is an approximating distribution represented using a neural network, we can differentiate the individual kernel terms in Equation 1 with respect to the simulated data $y_i$, and hence (via the chain rule) with respect to the parameters of the neural networks.

The MMD has been successfully used as a loss function in several generative adversarial networks. Dziugaite et al. (2015) and Li et al. (2015) propose training a feedforward neural network to minimize the MMD between simulations and data; Li et al. (2015) also propose minimizing the MMD between simulations and data that has been encoded using a pre-trained autoencoder. The generator in the MMD-GAN of Li et al. (2017) also aims to reduce the MMD between simulations and data, but learns the characteristic kernel in an adversarial manner by combining the kernel with a dynamically learned autoencoder.

## 3 WEIGHTED MMD

If we have unbiased samples from two distributions $\mathbb{P}$ and $\mathbb{Q}$, the estimator described in Equation 1 gives an unbiased estimate of the MMD between those two distributions. In a generative network context, we can therefore use this estimator as a loss function in order to modify the generator associated with $\mathbb{Q}$ so that the MMD between the two distributions is minimized.

However, this relies on having access to unbiased samples from our target distribution $\mathbb{P}$. In practice, our data may have been gathered using biased sampling practices: A dataset of images of faces may over-represent white faces; datasets generated from medical experiments may over-represent male patients; datasets generated from on-campus studies may over-represent college-aged students. If

our data is a biased sample of our target distribution, this estimator will estimate the difference between our simulations and the biased empirical distribution, so our simulations will recreate the biases therein.

In this section, we propose an estimator for the MMD between two distributions $\mathbb{P}$ and $\mathbb{Q}$ when we have access to $\mathbb{P}$ only via some biased sample selection mechanism. Concretely, we assume that $\mathbb{P}$ is our target of interest, but our observed data are actually sampled from a related distribution $T(x)\mathbb{P}(x)$. We can think of $T(x)$ as an appropriately scaled "thinning function" $\widetilde{T}(x) = ZT(x)$ that characterizes the sample selection mechanism. In other words, we assume that candidate samples $x^*$ are sampled from $\mathbb{P}$, and these candidates are selected into our pool with probability $\widetilde{T}(x^*)$; the normalizing constant $Z$ ensures that $T(x)\mathbb{P}(x) = \frac{\widetilde{T}(x)\mathbb{P}(x)}{Z}$ is a valid probability distribution. While in the remainder of this paper we will continue to use the language of biased sample selection mechanisms, we note that this framework can also be used if our data are unbiased samples but we want to explicitly modulate our target distribution via some function $F(x)$ so that we generate simulations from $F(x)\mathbb{P}(x)$; in this setting we can treat the transformed target as $\mathbb{P}$ and let $T(x) = 1/F(x)$.

For now, we assume that our thinning function $\widetilde{T}(x)$ is known; we discuss ways to estimate or specify it in Section 4. Our estimation problem becomes an importance sampling problem: We have samples from $T(\cdot)\mathbb{P}(\cdot)$, and we want to use these to estimate $\text{MMD}^2[\mathbb{P}, \mathbb{Q}]$, which is a function of the target distribution $\mathbb{P}$. Importance sampling provides a method for constructing an estimator for the expectation of a function $\phi(x)$ with respect to a distribution $\mathbb{P}$, by taking an appropriately weighted sum of evaluations of $\phi$ at values sampled from a different distribution $\mathbb{P}'$. If we knew the normalizing constant $Z$, we could construct an unbiased estimator of the MMD between $\mathbb{P}$ and $\mathbb{Q}$ by weighting each function evaluation associated with sample $x$ from $T(x)\mathbb{P}(x)$ with the likelihood ratio $\mathbb{P}(x)/T(x)\mathbb{P}(x) = 1/T(x)$, i.e.

$$\widehat{M_u} = \frac{1}{m(m-1)} \sum_{i=1}^{m} \sum_{j \neq i}^{m} \frac{1}{T(x_i)T(x_j)} k(x_i, x_j) + \frac{1}{n(n-1)} \sum_{i=1}^{n} \sum_{j \neq i}^{n} k(y_i, y_j)$$
$$- \frac{2}{mn} \sum_{i=1}^{m} \sum_{j=1}^{n} \frac{1}{T(x_i)} k(x_i, y_j). \tag{2}$$

However, the normalizing constant $Z$ depends on both $\widetilde{T}$ and $\mathbb{P}$. We will not, in general, know an analytic form for $\mathbb{P}$, so we will not be able to calculate $Z$. Since we will only be able to evaluate $T(x)$ up to a normalizing constant, we cannot work with Equation 2 directly. Instead, we can construct a biased estimator $\widehat{M_d}$ by using self-normalized importance weights,

$$\widehat{M_b} = \sum_{i=1}^{m} \sum_{j \neq i}^{m} w(x_i)w(x_j)k(x_i, x_j) + \frac{1}{n(n-1)} \sum_{i=1}^{n} \sum_{j \neq i}^{n} k(y_i, y_j) - \frac{2}{n} \sum_{i=1}^{m} \sum_{j=1}^{n} w(x_i)k(x_i, y_j) \tag{3}$$

where $w(x_i) = \frac{1/\widetilde{T}(x_i)}{\sum_{j=1}^{m} 1/\widetilde{T}(x_j)}$.

We refer to the estimator $\widehat{M_b}$ in Equation 3 as the weighted MMD estimator. While this estimator is biased due to the self-normalized weights, this bias will decrease as $1/m$ where $m$ is the number of samples from $T(\cdot)\mathbb{P}(\cdot)$. Further, this biased estimator will often have lower variance than the unbiased estimator in Equation 2 (Robert & Casella, 2004).

## 4 LEARNING THE THINNING FUNCTION $\widetilde{T}(x)$

In Section 3, we assumed that our target distribution $\mathbb{P}$ is only accessible via a biased sample selection mechanism characterized by a thinning function $\widetilde{T}(x)$, so that our samples are actually distributed (up to a normalizing constant) according to $\widetilde{T}(x)\mathbb{P}(x)$. If we know the thinning function $\widetilde{T}(x)$ that describes our sample selection mechanism, we can use Equation 3 directly in the loss function of our generator.

However, in practice we will not have access to this thinning function. Rather, we are likely to be in a situation where a practitioner has noticed that one or more classes is over- or under-represented, either in the dataset or in simulations from a generative network trained on the data. If our dataset is fully labeled, we could either manually re-balance our dataset to match the target distribution, or use these labels to estimate $\tilde{T}$ by comparing the number of data points in various subsets of $\mathcal{X}$ with the expected number under our target distribution.

Even if we do not have a fully labeled dataset, we might be able to label a subset of examples and use these to estimate $T(x)$. For example, assume we have an image dataset with more pictures of men than women, and that we select and label a random subset of this dataset. A reasonable estimate for our thinning function would be to set $T(\text{men}) = 1$ and approximate $T(\text{women})$ by the sample ratio of men to women in our labeled subset. In this simple two-class setting, to extrapolate values of $T$ across our observation space $X$ we can run a logistic regression using the labeled subset.

In a more complicated problem, we may need to deploy more sophisticated regression tools, but the problem remains one of function estimation from labeled data. Given a set of labeled exemplars and user-specified estimates of the thinning function evaluated at those exemplars (which could be based on demographic statistics or domain knowledge, or desired statistics of the simulation distribution), we could learn a thinning function $T(x)$ using techniques such as neural network function estimation or Gaussian process regression.

## 5 RELATED WORK

While it has received little attention in the deep learning literature, the problem of correcting for biased sample selection mechanisms is familiar in the field of survey statistics. Our estimator is related to inverse probability weighting (IPW), originally proposed by Horvitz & Thompson (1952) and still in wide use today (e.g. Mansournia & Altman, 2016). IPW is used in stratified random sampling study designs to correct parameter estimates for intentional bias in sampling. Under IPW, each data point is assigned a weight proportionate to the inverse of its selection probability, so that samples from classes which are disproportionately unlikely to be seen in the sample are assigned increased weights to counteract the undersampling of their classes. This is the same form as the weights in our sampling scheme, and serves a similar purpose, although it is not placed in the context of the MMD between two distributions and assumes that the probability of selection is known for each observation.

This work also follows increased awareness of the effects of biased data on machine learning outcomes, and interest in mitigating these effects. For example, Caliskan et al. (2017) and Bolukbasi et al. (2016) explore how biases and prejudices expressed in our language manifest in the word embeddings and representations found. Zhao et al. (2017b) find that popular image labeling datasets exhibit gender stereotypes and that algorithms trained on these datasets tend to amplify these stereotypes.

## 6 EVALUATION

We consider two experimental settings: One where the form of the sampling bias is known, and one where it is estimated from data.

### 6.1 CORRECTING FOR BIASED SAMPLE SELECTION WITH A KNOWN THINNING FUNCTION

In Section 3, we discussed how, for a given thinning function $\widetilde{T}(x)$, we can construct an estimator for the MMD between the distribution $\mathbb{Q}$ implied by our generator and the underlying *unbiased* data distribution $\mathbb{P}$. To demonstrate the efficacy of this in practice, we assume $\mathbb{P}$ to be a mixture of two Gaussians (Figure 1a), and let $\widetilde{T} = \frac{0.5}{1+\exp(10(x-1))} + 0.5$ be a scaled logistic function as shown in Figure 1b. The resulting data is distributed according to Figure 1c.

We construct a simple generator network taking univariate random noise as input, and consisting of six fully-connected layers, each with three nodes and exponential linear unit activations, and with a univariate output with no activation. We train this network using both the standard unbiased MMD

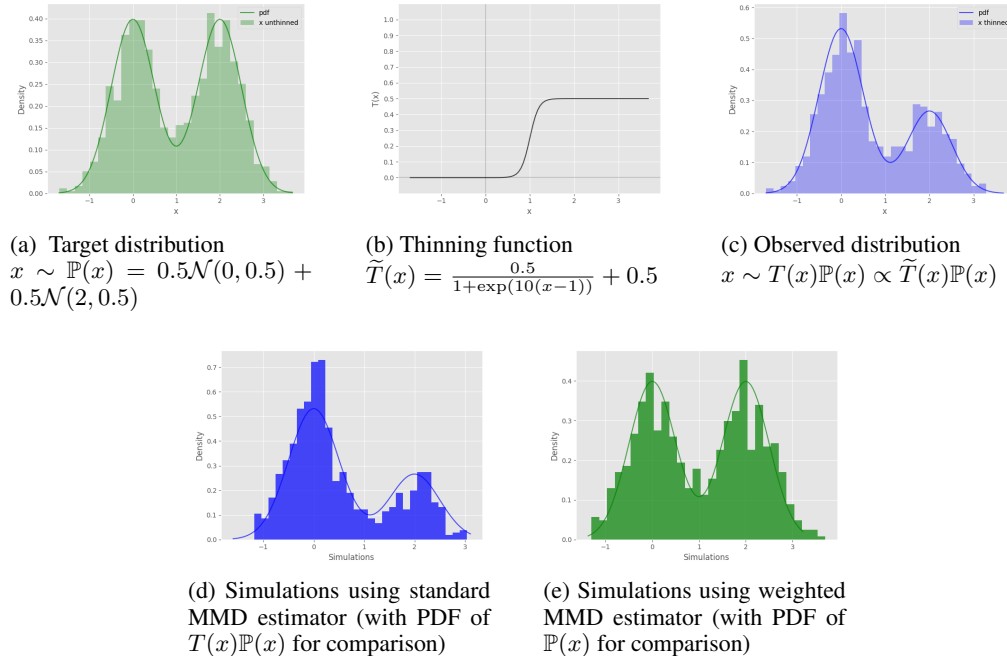

(a) Target distribution
$x \sim \mathbb{P}(x) = 0.5\mathcal{N}(0, 0.5) + 0.5\mathcal{N}(2, 0.5)$

(b) Thinning function
$\widetilde{T}(x) = \frac{0.5}{1+\exp(10(x-1))} + 0.5$

(c) Observed distribution
$x \sim T(x)\mathbb{P}(x) \propto \widetilde{T}(x)\mathbb{P}(x)$

(d) Simulations using standard MMD estimator (with PDF of $T(x)\mathbb{P}(x)$ for comparison)

(e) Simulations using weighted MMD estimator (with PDF of $\mathbb{P}(x)$ for comparison)

Figure 1: Output of GANs trained on data sampled according to $T(x)\mathbb{P}(x)$. Simulation plots PDFs are computed using NumPy's random normal sampler, mixed to create the appropriate mode ratios.

estimator of Equation 1, and the weighted MMD estimator proposed in Equation 3, using ADAM optimization with a learning rate of 0.001.

Figures 1e and 1d show histograms of samples generated using the two estimators. We see that, as expected, the standard MMD estimator does a good job of replicating the empirical distribution of the data presented to it. The weighted MMD estimator, however, is able to replicate the target distribution $\mathbb{P}$, even though it only has access to samples from $T(x)\mathbb{P}(x)$.

## 6.2 LEARNING AND CORRECTING FOR A BIASED SAMPLE SELECTION MECHANISM

In practice we are unlikely to know a functional form for $\widetilde{T}(x)$, the function that describes the form of the sampling bias. Here we consider the case where we must estimate this function from data. We consider as an example a dataset containing zeros and ones from the MNIST dataset. We assume our target distribution contains 50% zeros and 50% ones, but that due to a biased selection procedure our data contains 80% zeros and 20% ones.

We hypothesise that the practitioner using this dataset is aware that there is a discrepancy, but does not know how to translate this into a functional form for $\widetilde{T}$. Instead, the practitioner labels examples of each class, and supplies an estimate of how much each class is under-represented. In our example, we assume the practitioner has labeled 800 zeros and 200 ones, and estimates that ones are under-sampled according to a thinning probability of 0.75. We note that this number can easily be estimated from a data sample if the practitioner knows the global ratio of zeros and ones.

Table 1: Comparison of different estimators on the proportion of generated images classified as ones.

| Standard MMD | Weighted MMD (initialized to standard MMD) | Weighted MMD (random initialization) |
|---|---|---|
| $0.2201 \pm 0.0027$ | $0.3316 \pm 0.0028$ | $\mathbf{0.3888 \pm 0.0027}$ |

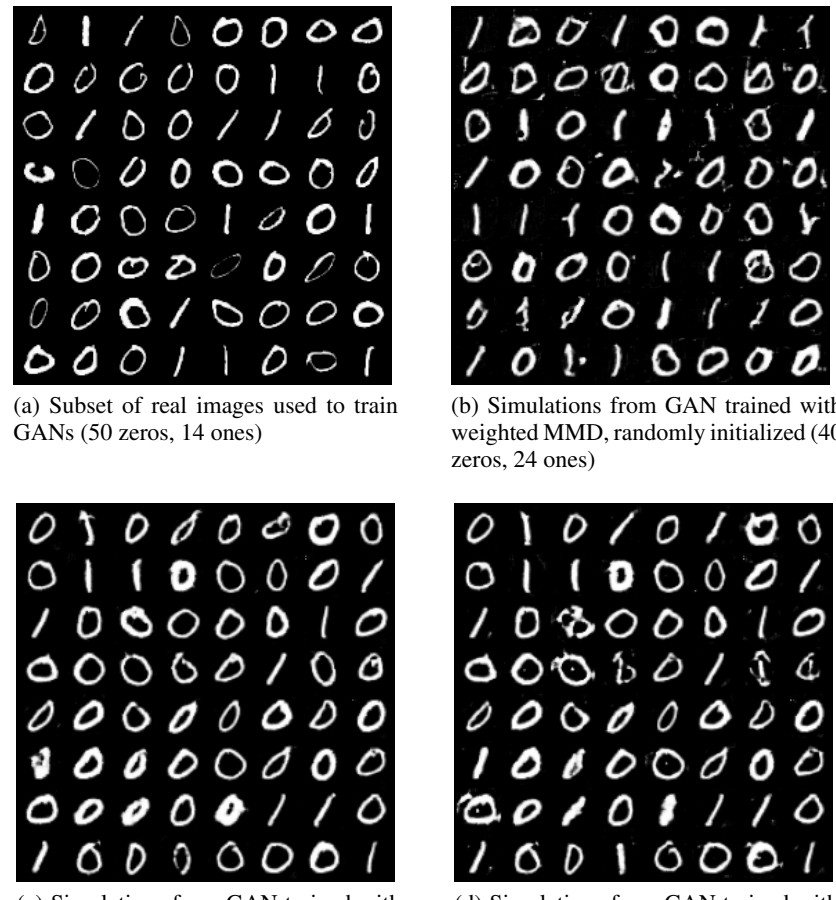

(a) Subset of real images used to train GANs (50 zeros, 14 ones)

(b) Simulations from GAN trained with weighted MMD, randomly initialized (40 zeros, 24 ones)

(c) Simulations from GAN trained with MMD (53 zeros, 11 ones)

(d) Simulations from GAN trained with weighted MMD, initialized to MMD (47 zeros, 17 ones)

Figure 2: Data used to train the GANs, and simulations from GANs trained using unweighted and weighted MMD.

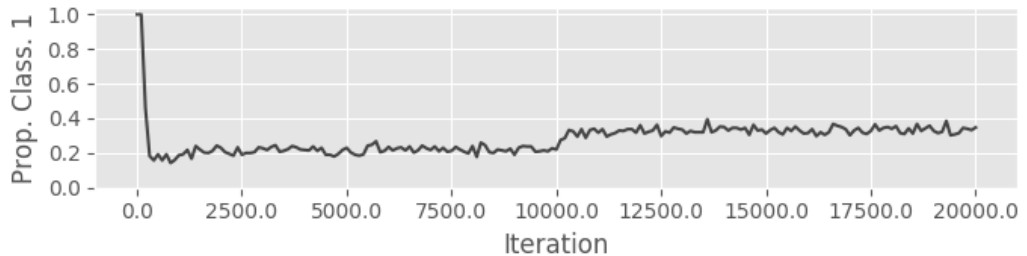

Figure 3: Proportion of images classified as ones, over time, for a network initially learned using the standard MMD estimator. The loss function switches to the weighted MMD estimator at iteration 10000, leading to an increase in the proportion of images classified as ones.

We use an architecture based on the MMD-GAN of Li et al. (2017), which incorporates an autoencoder, and trains a generator based on the MMD between the encoded representations of the data and of the simulations. The autoencoder is simultaneously learned in an adversarial manner, and serves to optimize the kernel used in MMD to maximally discriminate between the two distributions. Our experiments are run on an adapted version of the original MMD-GAN Torch code, which will be made available after publication.

Since the autoencoder provides a low-dimensional embedding of our images, we specify our thinning function $\widetilde{T}(x)$ on the space spanned by the encoder. After each update of the autoencoder, we estimate $\widetilde{T}$ with the labeled set, using appropriately scaled logistic regression. We then replace the standard MMD estimator in the loss function with our weighted estimator, calculating using the estimated $\widetilde{T}$.

One could imagine using the weighted estimator in two scenarios. We might be already aware of the bias in our data, in which case we would simply train our GAN using the weighted MMD estimator. Alternatively, we might only become aware of the bias after already training a GAN using the standard estimator, in which case we might initialize our network weights to the resulting pre-trained values. To simulate these scenarios, we trained two networks using the weighted estimator: one initialized to a pre-trained MMD-GAN, and one randomly initialized.

Figure 2a shows a random subset of the data used to train the GANs. Due to the biased sample selection method, there are significantly more zeros than ones. Figure 2c shows simulations generated by minimizing the standard MMD estimator using this biased data; as expected, it reflects the greater proportion of zeros compared to ones. By contrast, the simulations trained by minimizing the weighted MMD estimator, which estimates the MMD from the underlying target distribution as estimated by our thinning function, shows a marked increase in the number of ones, without any obvious difference in simulation quality. All networks were trained for 10,000 iterations.

To further quantify these results, we used the (unscaled) logistic regression used to specify the thinning function to classify the simulated images as zeros or ones. Table 1 shows the proportion of ones obtained using the three scenarios (standard MMD estimator, weighted MMD estimator initialized with a pre-trained network, and weighted MMD estimator with random initialization). We see that, as expected, the network trained using the standard MMD estimator produces around 20% ones, reflecting the proportion in the dataset. The networks trained using the weighted MMD estimator achieve a significantly higher percent of ones, with the randomly initialized network performing better than the pre-trained network.

While both the networks trained using the weighted MMD estimator simulate a higher proportion of ones than is present in our dataset, neither reaches the 50:50 ratio in our assumed target distribution. We believe this is due to inability of the network to fully converge to a solution that gives zero loss. This interpretation is supported by the fact that the randomly initialized network converges to a better solution than the pre-initialized network, despite the architecture and objective function being the same in both cases. To visualize the changes introduced by using the weighted MMD estimator, in Figure 3 we plot the proportion of ones on a network that is trained for 10,000 iterations using the standard MMD estimator, then for 10,000 iterations using the weighted MMD estimator. We see that the proportion quickly rises after the loss function is altered.

## 7 DISCUSSION AND FUTURE WORK

We have presented an asymptotically unbiased estimator for the MMD between two distributions $\mathbb{P}$ and $\mathbb{Q}$, for use when we only have access to $\mathbb{P}$ via a biased sampling mechanism. This mechanism can be specified by a known or estimated thinning function $\widetilde{T}(x)$, where samples are then assumed to come from a distribution $\widetilde{T}(x)\mathbb{P}(x)/Z$. We show that this estimator can be used to manipulate the distribution of simulations learned by a generative network, in order to correct for sampling bias or to explicitly change the distribution according to a user-specified function.

When the thinning function is unknown, it can be estimated from labeled data. We demonstrate this in an interpretable experiment using partially labeled images, where we jointly estimate the thinning function alongside the generator weights. An obvious next step is to explore the use of more sophisticated thinning functions appropriate for complex, multimodal settings.

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
