# OpenReview forum: "Directing Generative Networks with Weighted Maximum Mean Discrepancy"
_ICLR.cc/2018/Conference — Reject_

### Official Review · AnonReviewer1 · 2017-11-21
**Interesting idea, not enough evaluation**

**Rating:** 4
**Confidence:** 4

**Review:**

This paper proposes an importance-weighted estimator of the MMD, in order to estimate the MMD between distributions based on samples biased according to a known scheme. It then discusses how to estimate the scheme when it is unknown, and further proposes using it in either the MMD-based generative models of Y. Li et al. (2015) / Dziugaite et al. (2015), or in the MMD GAN of C.-L. Li et al. (2017).

The estimator itself is natural (and relatively obvious), though it has some drawbacks that aren't fully discussed (below).

The application to GAN-type learning is reasonable, and topical. The first, univariate, experiment shows that the scheme is at least plausible. But the second experiment, involving a simple T ratio based on whether an MNIST digit is a 0 or a 1, doesn't even really work! (The best model only gets the underrepresented class from 20% up to less than 40%, rather than the desired 50%, and the "more realistic" setting only to 33%.)

It would be helpful to debug whether this is due to the classifier being incorrect, estimator inaccuracies, or what. In particular, I would try using T based on a pretrained convnet independent of the autoencoder representation in the MMD GAN, to help diagnose where the failure mode comes from.

Without at least a working should-be-easy example like this, and with the rest of the paper's technical contribution so small, I just don't think this paper is ready for ICLR.

It's also worth noting that the equivalent algorithm for either vanilla GANs or Wasserstein GANs would be equally obvious.

Estimator:

In the discussion about (2): where does the 1/m bias come from? This doesn't seem to be in Robert and Casella section 3.3.2, which is the part of the book that I assume you're referring to (incidentally, you should specify that rather than just citing a 600-page textbook).

Moreover, it is worth noting that Robert and Cassela emphasize that if E[1 / \tilde T] is infinite, the importance sampling estimator can be quite bad (for example, the estimator may have infinite variance). This happens when \tilde T puts mass in a neighborhood around 0, i.e. when the thinned distribution doesn't have support at any place that P does. In the biased-observations case, this is in some sense unsurprising: if we don't see *any* data in a particular class of inputs, then our estimates can be quite bad (since we know nothing about a group of inputs that might strongly affect the results). In the modulating case, the equivalent situation is when F(x) lacks a mean, which seems less likely. Thus although this is probably not a huge problem for your case, it's worth at least mentioning. (See also the following relevant blog posts:  https://radfordneal.wordpress.com/2008/08/17/the-harmonic-mean-of-the-likelihood-worst-monte-carlo-method-ever/ and https://xianblog.wordpress.com/2012/03/12/is-vs-self-normalised-is/ .)

The paper might be improved by stating (and proving) a theorem with expressions for the rate of convergence of the estimator, and how they depend on T.


Minor:

Another piece of somewhat-related work is Xiong and Schneider, Learning from Point Sets with Observational Bias, UAI 2014.

Sutherland et al. 2016 and 2017, often referenced in the same block of citations, are the same paper.

On page 3, above (1): "Since we have projected the distributons into an infinite-dimensional space, the distance between the two distributions is zero if and only if all their moments are the same." An infinite-dimensional space isn't enough; the kernel must further be characteristic, as you mention. See e.g. Sriperumbuder et al. (AISTATS 2010) for more details.

Figure 1(b) seems to be plotting only the first term of \tilde T, without the + 0.5.

---

> ### Author Response · Authors · 2017-12-12
> **Thank you.**
>
> Thank you. You are correct about the proportion mismatch. While we move in the correct direction [amplifying the frequency of a target class in a partially labeled data set], we miss the desired theoretical distribution. We have identified some issues that may be contributing to this, and will include corrections in revised work. We agree that the thinning and weighting method can equivalently be applied in other GAN settings, and now see it as a method that applies to estimators in general. Your mention of failure modes and convergence is also appreciated, and will guide our future work.

---

### Official Review · AnonReviewer2 · 2017-11-27
**Unclear motivation and weak contribution!**

**Rating:** 4
**Confidence:** 4

**Review:**

This paper addresses the problem of sample selection bias in MMD-GANs. Instead of having access to an i.i.d. sample from the  distribution of interest, it is assumed that the dataset is subject to sample selection bias or the data has been gathered via a biased sample selection mechanism. Specifically, the observed data are drawn from the modified distribution T(x)P(x) where P(x) is the true distribution we aim to estimate and T(x) is an appropriately scaled "thinning function". Then, the authors proposed an estimate of the MMD between two distributions using weighted maximum mean discrepancy (MMD). The idea is in fact similar to an inverse probability weighting (IPW). They considered both when T(x) is known and when T(x) is unknown and must be estimated from the data. The proposed method was evaluated using both synthetic and real MNIST dataset.

In brief, sample selection bias is generally a challenging problem in science, statistics, and machine learning, so the topic of this paper is interesting. Nevertheless, the motivation for investigating this problem specifically in MMD-GANs is not clear. What motivated you to study this problem specifically for GAN in the first place? How does solving this problem help us understand or solve the sample selection bias in general? Will it shed light on how to improve the stability of GAN? Also, the experiment results are too weak to make any justified conclusion.

Some comments and questions:

- How is sample selection bias related to the stability issue of training GAN? Does it worsen the stability?
- Have estimators in Eq. (2) and Eq. (3) been studied before? Are there any theoretical guarantees that this estimate will convergence to the true MMD?
- On page 5, why T(men) = 1 and T(women) equals to the sample ratio of men to women in labeled subset?
- Can we use clustering to estimate the thinning function?

---

> ### Author Response · Authors · 2017-12-12
> **Thank you.**
>
> Thank you. You are correct that this thinning and weighting approach is applicable to any estimator under a biased sampling mechanism. We expect to broaden our discussion in a revision. Why GAN? We are excited about this approach in GANs because we recognized the issue of propagating bias in these generative models, and sought to correct it with a distributional discrepancy metric. We agree that more discussion of stability and convergence would strengthen the work, and are considering other thinning function techniques to make the model more applicable to practitioners. For example, in a large, unlabeled data setting, a practitioner could say, “Generate more like these 10 items.” We believe this will be a useful and theoretically well-founded adaptation for a variety of realistic data settings.

---

### Official Review · AnonReviewer3 · 2017-11-29
**Importance sampling correction to MMD for handling class imbalance**

**Rating:** 4
**Confidence:** 5

**Review:**

This paper presents a modification of the objective used to train generative networks with an MMD adversary (i.e. as in Dziugaite et al or Li et al 2015), where importance weighting is used to evaluate the MMD against a target distribution which differs from the data distribution. The goal is that this could be used to correct for known bias in the training data — the example considered here is for class imbalance for known, fixed classes.

Using importance sampling to estimate the MMD is straightforward only if the relationship between the data-generating distribution and the desired target distribution is somehow known and computable. Unfortunately the treatment of how this can be learned in general in section 4 is rather thin, and the only actual example here is on class imbalance. It would be good to see a comparison with other approaches for handling class imbalance. A straightforward one would be to use a stratified sampling scheme in selecting minibatches — i.e. rather than drawing minibatches uniformly from labeled data, select each minibatch by sampling an equal number of representatives from each class from the data. (Fundamentally, this requires explicit labels for whatever sort of bias we wish to correct for, for every entry in the dataset.) I don't think the demonstration of how to compute the MMD with an importance sampling estimate is a sufficient contribution on its own.

Also, I am afraid I do not understand the description of subfigures a through c in figure 1. The target distribution p(x) is given in 1(a), a thinning function in 1(b), and an observed distribution in 1(c). As described, the observed data distribution in 1(c) should be found by multiplying the density in 1(a) by the function in 1(b) and then normalizing. However, the function \tilde T(x) in 1(b) takes values near zero when x < 0, meaning the product \tilde T(x)p(x) should also be near zero. But in figure 1(c), the mode of p(x) near x=0 actually has higher probability than the mode near x=2, despite the fact that there \tilde T(x) \approx 0.5. I think this might simply be a mistake in the definition of \tilde T(x), and that rather it should be 1.0 - \tilde T(x), but in any case this is quite confusing.

I also am confused by the results in figure 2. I would have thought that the right column, where the thinning function is used to correct for the class imbalance, would then have approximately equal numbers of zeros and ones in the generative samples. But, there are still more zeros by a factor of around 2.

Minor note: please double-check references, there seem to be some issues; for example, Sutherland et al is cited twice, once as appearing at ICML 2016 and once as appearing at ICML 2017.

---

> ### Author Response · Authors · 2017-12-12
> **Thank you.**
>
> Thank you. Class imbalance in a large-scale, unlabeled data setting remains an important and common problem, which we hope this will address. We agree that our first approach at a thinning function is simplistic, and are considering a more complex classifier, for an active-learning approach, where a practitioner could say, “The generated set needs more like these 10”. Also, you are correct about the figures: in the first case, the function is correct but the image was wrong, and we noticed after submission; in the second case, we have since identified some issues that may contribute to missing the desired final distribution. This work demonstrated the correct initial behavior, and your comments will help us revise to meet those desired theoretical outcomes.

---

### Decision · Program_Chairs · 2018-01-29
**ICLR 2018 Conference Acceptance Decision**

**Decision:**

Reject

**Comment:**

The reviewers agree that the problem being addressed is interesting, however there are concerns with novelty and with the experimental results. An experiment beyond dealing with class imbalance would help strengthen this paper, as would experiments with other kinds of GANs.